# Learning Affordance Landscapes for Interaction Exploration in 3D Environments

**Tushar Nagarajan**
UT Austin and Facebook AI Research
tushar@cs.utexas.edu

**Kristen Grauman**
UT Austin and Facebook AI Research
grauman@fb.com

## Abstract

Embodied agents operating in human spaces must be able to master how their environment works: what objects can the agent use, and how can it use them? We introduce a reinforcement learning approach for *exploration for interaction*, whereby an embodied agent autonomously discovers the affordance landscape of a new unmapped 3D environment (such as an unfamiliar kitchen). Given an egocentric RGB-D camera and a high-level action space, the agent is rewarded for maximizing successful interactions while simultaneously training an image-based affordance segmentation model. The former yields a policy for acting efficiently in new environments to prepare for downstream interaction tasks, while the latter yields a convolutional neural network that maps image regions to the likelihood they permit each action, densifying the rewards for exploration. We demonstrate our idea with AI2-iTHOR. The results show agents can learn how to use new home environments intelligently and that it prepares them to rapidly address various downstream tasks like "find a knife and put it in the drawer." Project page: http://vision.cs.utexas.edu/projects/interaction-exploration/

## 1   Introduction

The ability to interact with the environment is an essential skill for embodied agents operating in human spaces. Interaction gives agents the capacity to modify their environment, allowing them to move from semantic navigation tasks (e.g., "go to the kitchen; find the coffee cup") towards complex tasks involving interactions with their surroundings (e.g., "heat some coffee and bring it to me").

Today's embodied agents are typically trained to perform specific interactions in a supervised manner. For example, an agent learns to navigate to specified objects [18], a dexterous hand learns to solve a Rubik's cube [4], a robot learns to manipulate a rope [40]. In these cases and many others, it is known a priori *what objects are relevant for the interactions and what the goal of the interaction is*, whether expressed through expert demonstrations or a reward crafted to elicit the desired behavior. Despite exciting results, the resulting agents remain specialized to the target interactions and objects for which they were taught.

In contrast, we envision embodied agents that can enter a novel 3D environment, move around to encounter new objects, and autonomously discern the affordance landscape—what are the interactable objects, what actions are relevant to use them, and under what conditions will these interactions succeed? Such an agent could then enter a new kitchen (say), and be primed to address tasks like "wash my coffee cup in the sink." These capabilities would mimic humans' ability to efficiently discover the functionality of even unfamiliar objects though a mixture of learned visual priors and exploratory manipulation.

To this end, we introduce the *exploration for interaction* problem: a mobile agent in a 3D environment must autonomously discover the objects with which it can physically interact, and what actions are valid as interactions with them.

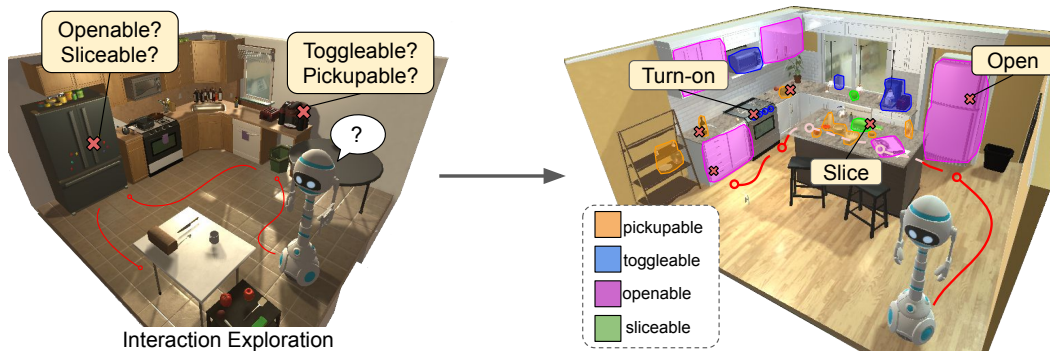

**Figure 1: Main idea.** We train *interaction exploration* agents to quickly discover what objects can be used and how to use them. Given a new, unseen environment, our agent can infer its visual affordance landscape, and efficiently interact with all the objects present. The resulting exploration policy and affordance model prepare the agent for downstream tasks that involve multiple object interactions.

Exploring for interaction presents a challenging search problem over the product of all objects, actions, agent positions, and action histories. Furthermore, many objects are hidden (e.g., in drawers) and need to be discovered, and their interaction dynamics are not straightforward (e.g., cannot *open* an already opened door, can only *slice* an apple if a knife is *picked up*). In contrast, exploration for navigating a static environment involves relatively small action spaces and dynamics governed solely by the presence/absence of obstacles [12, 50, 51, 18, 11, 47].

Towards addressing these challenges, we propose a deep reinforcement learning (RL) approach in which the agent discovers the affordance landscape of a new, unmapped 3D environment. The result is a strong prior for where to explore and what interactions to try. Specifically, we consider an agent equipped with an egocentric RGB-D camera and an action space comprised of navigation and manipulation actions (turn left, open, toggle, etc.), whose effects are initially unknown to the agent. We reward the agent for quickly interacting with all objects in an environment. In parallel, we train an affordance model online to segment images according to the likelihoods for each of the agent's actions succeeding there, using the partially observed interaction data generated by the exploration policy. The two models work in concert to functionally explore the environment. See Figure 1.

Our experiments with AI2-iTHOR [29] demonstrate the advantages of interaction exploration. Our agents can quickly seek out new objects to interact with in new environments, matching the performance of the best exploration method in 42% fewer timesteps and surpassing them to discover $1.33\times$ more interactions when fully trained. Further, we show our agent and affordance model help train multi-step interaction policies (e.g., washing objects at a sink), improving success rates by up to 16% on various tasks, with fewer training samples, despite sparse rewards and no human demonstrations.

## 2 Related Work

**Visual affordances** An affordance is the potential for action [22]. In computer vision, visual affordances are explored in various forms: predicting where to grasp an object from images and video [31, 32, 64, 38, 19, 62, 15, 5], inferring how people might use a space [48, 39] or tool [65], and priors for human body poses [26, 52, 58, 17]. Our work offers a new perspective on learning visual affordances. Rather than learn them passively from a static dataset, the proposed agent actively seeks new affordances via exploratory interactions with a dynamic environment. Furthermore, unlike prior work, our approach yields not just an image model, but also a *policy* for exploring interactions, which we show accelerates learning new downstream tasks for an embodied agent.

**Exploration for navigation in 3D environments** Recent embodied AI work in 3D simulators [36, 56, 60, 10] tackles navigation: the agent moves intelligently in an unmapped but static environment to reach a goal (e.g., [12, 11, 36, 6]). Exploration policies for visual navigation efficiently map the environment in an unsupervised "preview" stage [12, 50, 18, 11, 47, 46]. The agent is rewarded for maximizing the area covered in its inferred occupancy map [12, 11, 18], the novelty of the states visited [51], pushing the frontier of explored areas [46], and related metrics [47]. For a game setting in VizDoom, classic frontier-based exploration is improved by learning the visual appearance of hazardous regions (e.g., enemies, lava) where the agent's health score has previously declined [46].

In contrast to all the above, we study the problem of exploration for *interaction* in dynamic environments where the agent can modify the environment state (open/close doors, pick up objects etc.). Our

end goal is not to build a top-down occupancy map, but rather to quickly interact with as many objects as possible in a new environment. In other words, whereas exploration for navigation promotes rapidly completing a static environment map, exploration for interaction promotes rapidly completing the agent's understanding of its interactions in a dynamic environment.

**Interaction in 3D environments**   Beyond navigation, recent work leverages simulated interaction-based environments [21, 29, 55, 45] to develop agents that can also perform actions (e.g., moving objects, opening doors) with the goal of eventually translating policies to real robots [2, 1]. These tasks include answering questions ("how many apples are in the fridge?") that may require navigation [16] as well as interaction [25]. Towards service robotics, goal driven planning [63], instruction following [55], and cooking [21] agents are trained using imitation learning on expert trajectories.

Our idea to efficiently *explore* interactions is complementary. Rather than learn a task-specific policy from demonstrations, our approach learns task-agnostic exploration behavior from experience to quickly discover the affordance landscape. Our model can be coupled with a downstream task like those tackled above to accelerate their training, as we demonstrate in the experiments.

**Self-supervised interaction learning**   Prior work studies actively learning manipulation policies through self-supervised training for grasping [44, 42, 33, 35, 61], pushing/poking [3, 41] and drone control [20]. Unstructured *play* data has also been used to learn subgoal policies [34], which are then sampled to solve complex tasks. Object affordance models are learned for simple objects in table-top environments [23, 24] and for block pushing tasks in gridworlds [28]. We share the general idea of learning through interaction; however, we focus on high-level interaction policies requiring both navigation and manipulation (e.g., moving to the counter and picking up knife) rather than fine-grained manipulation policies (e.g., altering joint angles).

**Intrinsic motivation**   In the absence of external rewards from the environment, reinforcement learning agents can nonetheless focus their behavior to satisfy *intrinsic* drives [53]. Recent work formulates intrinsic motivation based on curiosity [43, 9, 27], novelty [51, 7], and empowerment [37] to improve video game playing agents (e.g., VizDoom, Super Mario) or increase object attention [27]. Our idea can be seen as a distinct form of intrinsic motivation, where the agent is driven to experience more interactions in the environment. Also, we focus on realistic human-centered 3D environments, rather than video games, and with high-level interactions that can change object state, rather than low-level physical manipulations.

## 3   Approach

Our goal is to train an interaction exploration agent to enter a new, unseen environment and successfully interact with all objects present. This involves identifying the objects that are interactable, learning to navigate to them, and discovering all valid interactions with them (e.g., discovering that the agent can *toggle* a light switch, but not a knife).

To address the challenges of a large search space and complex interaction dynamics, our agent learns visual affordances to help it intelligently select regions of the environment to explore and interactions to try. Critically, our agent builds this affordance model through its own experience interacting with the environment during exploration. For example, by successfully *opening* a cupboard, the agent learns that objects with handles are likely to be "openable". Our method yields an interaction exploration policy that can quickly perform object interactions in new environments, as well as a visual affordance model that captures where each action is likely to succeed in the egocentric view.

In the following, we first define the interaction exploration task (Sec. 3.1). Then, we show how an agent can train an affordance model via interaction experience (Sec. 3.2). Finally, we present our policy learning architecture that integrates interaction exploration and affordance learning, and allows transfer to goal-driven policy learning (Sec. 3.3).

### 3.1   Learning exploration policies for interaction

We want to train an agent to interact with as many objects as possible in a new environment. Agents can perform actions from a set $\mathcal{A} = \mathcal{A}_N \bigcup \mathcal{A}_I$, consisting of navigation actions $\mathcal{A}_N$ (e.g., move forward, turn left/right) and object interactions $\mathcal{A}_I$ (e.g., take/put, open/close).

The interaction exploration task is set up as a partially observable Markov decision process. The agent is spawned at an initial state $s_0$. At each time step $t$, the agent in state $s_t$ receives an observation

$(x_t, \theta_t)$ consisting of the RGB image $x_t$ and the agent's odometry[1] $\theta_t$, executes an action $a_t \sim \mathcal{A}$ and receives a reward $r_t \sim \mathcal{R}(s_t, a_t, s_{t+1})$. A recurrent network encodes the agent's observation history over time to arrive at the state representation. The agent is rewarded for each successful interaction with a new object $o_t$:

$$R(s_t, a_t, s_{t+1}) = \begin{cases} 1 & \text{if } a_t \in \mathcal{A}_I \text{ and } c(a_t, o_t) = 0 \\ 0 & \text{otherwise,} \end{cases} \quad (1)$$

where $c(a, o)$ counts how many times interaction $(a, o)$ has successfully occurred in the past. The goal is to learn an exploration policy $\pi_E$ that maximizes this reward over an episode of length $T$. See Sec. 3.3 for the policy architecture. The hard, count-based reward formulation only rewards the agent once per interaction, incentivizing broad coverage of interactions, rather than mastery of a few, which is useful for downstream tasks involving arbitrary interactions.

## 3.2 Affordance learning via interaction exploration

As the agent explores, it attempts interactions at various locations, only some of which succeed. These attempts partially reveal the *affordances* of objects — what interactions are possible with them — which we capture in a visual affordance model. An explicit model of affordances helps the agent decide what regions to visit (e.g., most interactions fail at walls, so avoid them) and helps extrapolate possible interactions with unvisited objects (e.g., opening one cupboard suggests that other handles are "openable"), leading to more efficient exploration policies.

At a high level, we train an affordance segmentation model $\mathcal{F}_A$ to transform an input RGB-D image into a $|\mathcal{A}_I|$-channel segmentation map, where each channel is a $H \times W$ map over the image indicating regions where a particular interaction is likely to succeed. Training samples for this model comes from the agent's interaction with the environment. For example, if it successfully picks up a kettle, pixels around that kettle are labeled "pickup-able", and these labels are propagated to all frames where the kettle is visible (both before and after the interaction took place), such that affordances will be recognizable even from far away. See Fig. 2 (right panel).

Specifically, for a trajectory $\tau = \{(s_t, a_t)\}_{t=1..T} \sim \pi_E$ sampled from our exploration policy, we identify time steps $t_1...t_N$ where interactions occur ($a_t \in \mathcal{A}_I$). For each interaction, the world location $p_t$ at the center of the agent's field of view is calculated by inverse perspective projection and stored along with the interaction type $a_t$ and success of the interaction $z_t$ in memory as $\mathcal{M} = \{(p_t, a_t, z_t)\}_{t=t_1..t_N}$. This corresponds to "marking" the target of the interaction.

At the end of the episode, for each frame $x$ in the trajectory, we generate a corresponding segmentation mask $y$ that highlights the position of all markers from any action that are visible in $x$. For each interaction $a_k$, the label for each pixel in the $k$-th segmentation mask slice $y^k$ is calculated as:

$$y_{ij}^k = \begin{cases} 0 & \text{if } \min_{(p,a,z) \in \mathcal{M}_k} d(p_{ij}, p) < \delta \text{ and } z = 0 \\ 1 & \text{if } \min_{(p,a,z) \in \mathcal{M}_k} d(p_{ij}, p) < \delta \text{ and } z = 1 \\ -1 & \text{otherwise} \end{cases} \quad (2)$$

where $\mathcal{M}_k \subseteq \mathcal{M}$ is the subset of markers corresponding to interaction $a_k$, $p_{ij}$ is the world location at that pixel, $d$ is euclidean distance, and $\delta$ is a fixed distance threshold (20cm). In other words, each pixel is labeled 0 or 1 for affordance $k$ depending on whether any marker has been placed nearby (within distance $\delta$) at any time along the trajectory, and is visible in the current frame. If no markers are placed, the pixel is labeled $-1$ for unknown. See Fig. 2 (right panel). This results in a $|\mathcal{A}_I| \times H \times W$ dimension segmentation label mask per frame, which we use to train $\mathcal{F}_A$.

These labels are sparse and noisy, as an interaction may fail with an object despite being valid in other conditions (e.g., opening an already opened cupboard). To account for this, we train two distinct segmentation heads using these labels to minimize a combination of cross entropy losses:

$$\mathcal{L}(\hat{y}_A, \hat{y}_I, y) = \mathcal{L}_{ce}(\hat{y}_A, y, \forall y_{ij} \neq -1) + \mathcal{L}_{ce}(\hat{y}_I, \mathbb{1}[y = -1], \forall y_{ij}) \quad (3)$$

where $\mathbb{1}[.]$ is the indicator function over labels. $\mathcal{L}_{ce}$ is standard cross entropy loss, but is evaluated over a subset of pixels specified by the third argument. Classifier output $\hat{y}_A$ scores whether each interaction is successful at a location, while $\hat{y}_I$ scores general interactibility ($y = -1$ vs. $y \neq -1$). The latter acts as a measure of uncertainty to ignore regions where markers are rarely placed, regardless of success (e.g., the ceiling, windows). The final score output by $\mathcal{F}_A$ is the product $\hat{y} = \hat{y}_A \times (1 - \hat{y}_I)$.

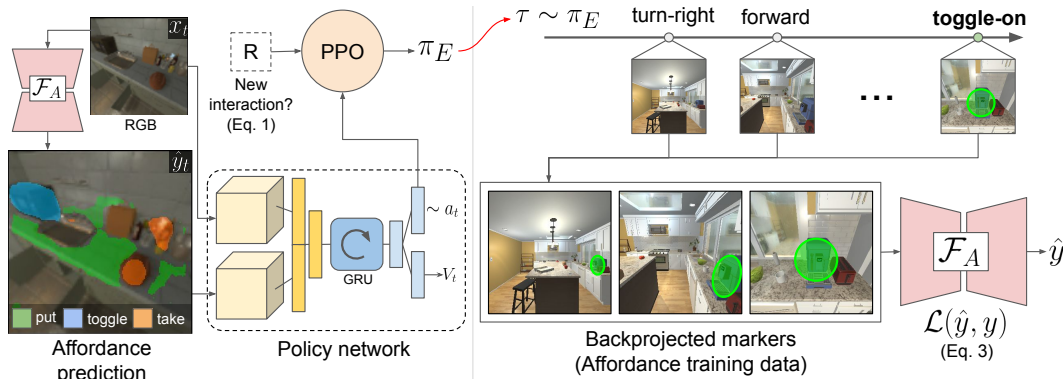

**Figure 2: Interaction exploration framework. Left panel:** Our policy network takes the current frame $x_t$ and predicted affordance maps $\mathcal{F}_A(x_t)$ as input to train a policy $\pi_E$ to maximize the interaction exploration reward in Equation 1 (Sec. 3.1). **Right panel:** As the policy network trains, trajectories sampled from $\pi_E$ are used to create affordance training samples to improve $\mathcal{F}_A$, by "marking" target locations of interactions and propagating these regions to other frames along the trajectory where the target is visible (green regions) (Sec. 3.2).

In our experiments, we consider two variants: one that marks interactions with a single point, and one that marks all points on the target object of the interaction. The former translates to fixed scale labels at the exact interaction location, supposing no prior knowledge about object segmentation. The latter is more general and considers the whole object as "interactable", leading to denser labels. In both cases, the object class and valid interactions are unknown to the agent.

### 3.3 Policy learning architecture and transfer

Next we put together both pieces—the interaction exploration objective and the affordance segmentations—in our policy learning framework. We adopt an actor-critic policy model and a U-Net [49] architecture for affordances. At each time step, we receive the current egocentric frame $x$ and generate its affordance maps $\hat{y} = \mathcal{F}_A(x)$. The visual observations and affordance maps are encoded using a 3-layer convolutional neural network (CNN) each, and then concatenated and merged using a fully connected layer. This is then fed to a gated recurrent unit (GRU) recurrent neural network to aggregate observations over time, and finally to an actor-critic network (fully connected layers) to generate the next action distribution and value. We train this network using PPO [54] for 1M frames, with rollouts of $T = 256$ time steps. See Fig. 2 (left) and Supp for architecture details.

We train the policy network and the segmentation model iteratively. As the agent explores, we store episodes drawn from the exploration policy, and create an affordance segmentation dataset as per Sec. 3.2. We train the affordance model using this dataset, and use the updated model to generate $\hat{y}$ to further train the policy network described above. See Supp for training schedule.

The result of this process is an interaction exploration policy $\pi_E$ that can quickly master object interactions in new environments, as well as a visual affordance model $\mathcal{F}_A$, which captures where interactions will likely succeed in the current view. In addition, we show the policy transfers to better learn downstream tasks. Specifically, we freeze the weights of the policy network and $\mathcal{F}_A$, and fine-tune only the actor-critic linear layers using the downstream task's reward (cf. Sec. 4.2).

## 4 Experiments

We evaluate agents' ability to interact with as many objects as possible (Sec. 4.1) and enhance policy learning on downstream tasks (Sec. 4.2).

**Simulation environment** We experiment with AI2-iTHOR [30] (see Fig. 1), since it supports context-specific interactions that can change object states, vs. simple physics-based interactions in other 3D indoor environments [59, 8]. We use all kitchen scenes; kitchens are a valuable domain since many diverse interactions with objects are possible, as also emphasized in prior work [14, 38, 21]. The scenes contain objects from 69 classes, each of which supports 1-5 interactions. We split the 30 scenes into training (20), validation (5), and testing (5) sets. We randomize objects' positions and states (isOpen, isToggled etc), agent start location, and camera viewpoint when sampling episodes.

| | Take | | Put | | Open | | Close | | Turn-on | | Turn-off | | Slice | | Average | |
|---|---|---|---|---|---|---|---|---|---|---|---|---|---|---|---|---|
| | Prec | Cov | Prec | Cov | Prec | Cov | Prec | Cov | Prec | Cov | Prec | Cov | Prec | Cov | Prec | Cov |
| RANDOM | 2.61 | 9.46 | 2.30 | 19.25 | 5.51 | 14.80 | 3.89 | 9.20 | 1.46 | 9.48 | 0.92 | 5.74 | 0.02 | **0.44** | 2.39 | 9.81 |
| RANDOM+ | 3.17 | 10.87 | 3.01 | 19.28 | 7.66 | 16.67 | 7.05 | 14.60 | 1.81 | 10.35 | 2.00 | 11.33 | 0.01 | 0.10 | 3.53 | 11.89 |
| CURIOSITY [43] | 2.41 | 9.33 | 2.12 | 19.75 | 5.25 | 14.26 | 3.62 | 8.94 | 1.53 | 10.47 | 0.88 | 6.01 | 0.00 | 0.10 | 2.26 | 9.84 |
| NOVELTY [51] | 3.09 | 11.95 | 3.11 | 24.62 | 8.47 | 22.82 | 6.09 | 20.53 | 1.52 | 10.60 | 1.47 | 10.97 | 0.00 | 0.00 | 3.39 | 14.50 |
| OBJCOVERAGE [47] | 5.60 | 15.54 | 5.36 | 30.29 | 6.34 | 21.16 | 5.66 | 18.88 | 3.04 | 15.56 | 3.07 | 15.93 | 0.00 | 0.00 | 4.15 | 16.77 |
| INTEXP | **8.53** | **19.23** | **6.35** | **39.62** | **10.18** | **30.05** | **9.82** | **24.43** | **12.42** | **23.19** | **11.91** | **18.45** | **0.04** | 0.06 | **8.46** | **22.15** |

**Table 1: Exploration performance per interaction.** Our policy is both more precise (prec) and discovers more interactions (cov) than all other methods. Methods that cycle through actions eventually succeed, but at the cost of interaction failures along the way.

Agents can both navigate: $\mathcal{A}_N = \{$move forward, turn left/right $30°$, look up/down $15°\}$, and perform interactions with objects in the center of the agent's view: $\mathcal{A}_I = \{$take, put, open, close, toggle-on, toggle-off, slice$\}$. While the simulator knows what actions are valid given where the agent is, what it is holding, and what objects are nearby, all this knowledge is hidden from the agent, who only knows if an action succeeds or fails.

**Baselines** We compare several methods:

- **RANDOM** selects actions uniformly at random. **RANDOM+** selects random navigation actions from $\mathcal{A}_N$ to reach unvisited locations, then cycles through all possible object interactions in $\mathcal{A}_I$.
- **CURIOSITY** [9, 43] rewards actions that lead to states the agent cannot predict well.
- **NOVELTY** [57, 51, 7] rewards visits to new, unexplored physical locations. We augment this baseline to cycle through all interactions upon reaching a novel location.
- **OBJCOVERAGE** [18, 47] rewards an agent for visiting new objects (moving close to it, and centering it in view), but not for interacting with them. We similarly augment this to cycle over all interactions.
  The above three are standard paradigms for exploration. See Supp for details.

**Ablations** We examine several variants of the proposed interaction exploration agent. All variants are rewarded for interactions with novel objects (Equation 1) and use the same architecture (Sec. 3.3).

- **INTEXP(RGB)** uses only the egocentric RGB frames to learn the policy, no affordance map.
- **INTEXP(SAL)** uses RGB plus heatmaps from a pretrained saliency model [13] as input, which highlight salient objects but are devoid of affordance cues.
- **INTEXP(GT)** uses ground truth affordances from the simulator.
- **INTEXP(PT)** and **INTEXP(OBJ)** use affordances learned on-the-fly from interaction with the environment by marking fixed sized points or whole objects, respectively (see Sec. 3.2). INTEXP(PT) is our default model for experiments unless specified.

In short, RANDOM and RANDOM+ test if a learned policy is required at all, given small and easy to navigate environments. NOVELTY, CURIOSITY, and OBJCOVERAGE test whether intelligent interaction policies fall out naturally from traditional exploration methods. Finally, the interaction exploration ablations test how influential learned visual affordances are in driving interaction discovery.

### 4.1 Affordance driven interaction exploration

First we evaluate how well an agent can locate and interact with all objects in a new environment.

**Metrics.** For each test environment, we generate 80 randomized episodes of 1024 time steps each. We create an "oracle" agent that takes the shortest path to the next closest object and performs all valid interactions with it, to gauge the maximum number of possible interactions. We report (1) Coverage: the fraction of the maximum number of interactions possible that the agent successfully performs and (2) Precision: the fraction of interactions that the agent attempted that were successful.

**Interaction exploration.** Fig. 3 (left) shows interaction coverage on new, unseen environments over time, averaged over all episodes and environments. See Supp for environment-specific results. Even though CURIOSITY is trained to seek hard-to-predict states, like the non-trained baselines it risks performing actions that block further interaction (e.g., opening cupboards blocks paths). RANDOM+, NOVELTY, and OBJCOVERAGE seek out new locations/objects but can only cycle through all interactions, leading to slow discovery of new interactions.

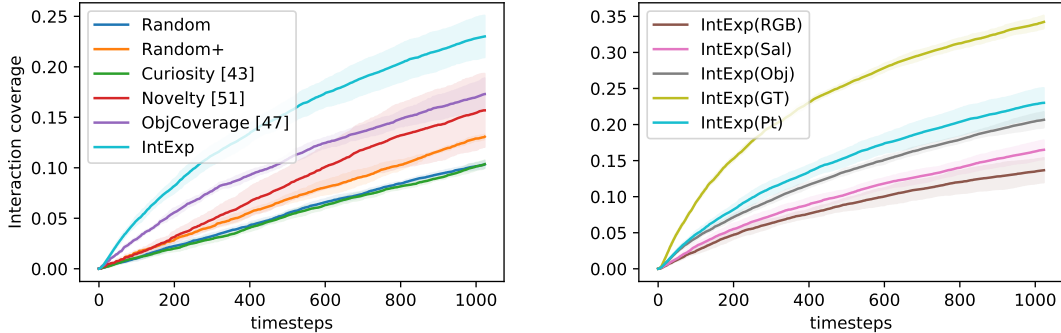

Figure 3: **Interactions discovered vs. time on unseen environments. Left:** Our agents discover the most object interactions, and do so faster than all other methods, especially early on (T<256). **Right:** In the ablation, models that learn an affordance model to guide exploration outperform those with weaker priors (like saliency), and come closest to the model with access to ground truth (GT) affordances. Results are from three training runs.

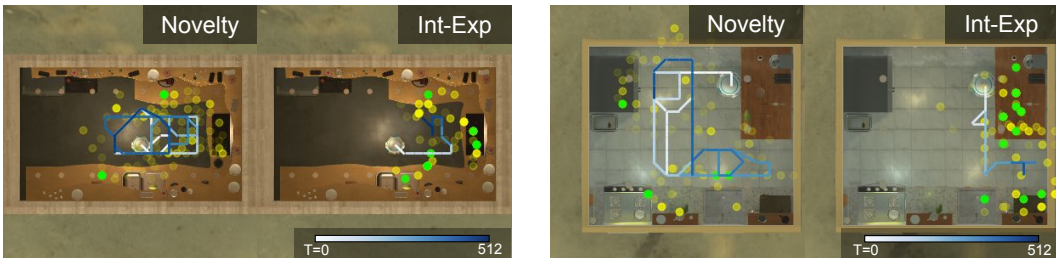

Figure 4: **Interaction policy examples on test environments.** Green dots are successfully discovered interactions, yellow are all interaction attempts. NOVELTY visits many parts of the space, but fails to intelligently select which actions to attempt. Our policy learns to visit only relevant locations to interact with objects.

Our full model with learned affordance maps leads to the best interaction exploration policies, and discovers 1.33× more unique object interactions than the strongest baseline. Moreover, it performs these interactions quickly — it discovers the same number of interactions as RANDOM+ in 63% fewer time-steps. Our method discovers 2.5× more interactions than NOVELTY at $T$=256.

Fig. 3 (right) shows variants of our method that use different visual priors. INT-EXP(RGB) has no explicit RoI model and performs worst. In INT-EXP(SAL), saliency helps distinguish between objects and walls/ceiling, but does not reveal what interactions are possible with salient objects as our affordance model does. INTEXP(OBJ) performs well during training — 0.236 vs. 0.252 coverage compared to INTEXP(PT) — but suffers more from noisy marker labels as it trains using whole object masks. INTEXP(PT) marks exact target locations and generalizes better to unseen environments, but yields more conservative affordance predictions (see Fig. 5).

Table 1 shows an action-wise breakdown of coverage and precision. In general, many objects can be opened/closed (drawers, fridges, kettles etc.) resulting in more instances covered for those actions. All methods rarely slice objects successfully as it requires first locating and picking up a knife (all have cov <1%). This requires multiple steps that are unlikely to occur randomly, and so is overlooked by trained agents in favor of more accessible objects/interactions. Importantly, methods that cycle through actions eventually interact with objects, leading to moderate coverage, but very low precision since they do not know how to prioritize interactions. This is further exemplified in Fig. 4. NOVELTY tends to seek out new locations, regardless of their potential for interaction, resulting in few successes (green dots) and several failed attempts (yellow dots). Our agent selectively navigates to regions with objects that have potential for interaction. See Supp for more examples.

**Affordance prediction.** In addition to exploration policies, our method learns an affordance model. Fig. 5 evaluates the INTEXP agents for reconstructing the ground truth affordance landscape of 23,637 uniformly sampled views from unseen test environments. We report mean average precision over all interaction classes. The ALL-ONES baseline assigns equal scores to all pixels. INTEXP(SAL) simply repeats its saliency map $|\mathcal{A}_I|$ times as the affordance map. Other agents from Fig. 3 do not train affordance models, thus cannot be compared. Our affordance models learn maps tied to the individual actions of the exploring agent and result in the best performance.

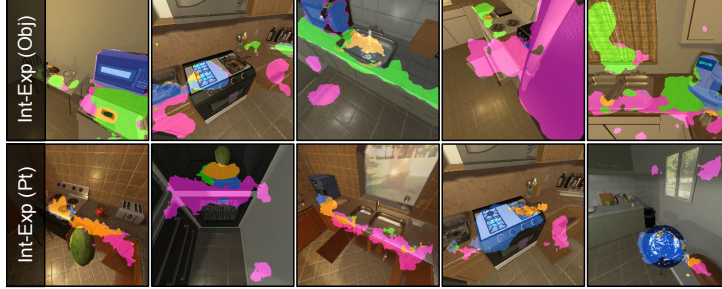

| | mAP |
|---|---|
| ALL-ONES | 8.9 |
| INTEXP(SAL) | 13.4 |
| INTEXP(PT) | 16.3 |
| INTEXP(OBJ) | **26.5** |
| INTEXP(GT) | 37.3 |

**Figure 5: Affordance prediction results** ( ■ take ■ put ■ open ■ toggle ). Our models are trained with masks automatically inferred via exploration — it does not have access to ground truth affordances. Last column shows failure cases (curtain, pan) due to noisy/incomplete interaction data.

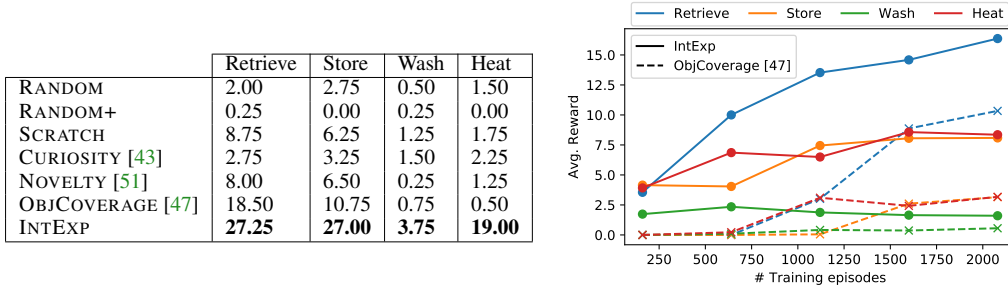

| | Retrieve | Store | Wash | Heat |
|---|---|---|---|---|
| RANDOM | 2.00 | 2.75 | 0.50 | 1.50 |
| RANDOM+ | 0.25 | 0.00 | 0.25 | 0.00 |
| SCRATCH | 8.75 | 6.25 | 1.25 | 1.75 |
| CURIOSITY [43] | 2.75 | 3.25 | 1.50 | 2.25 |
| NOVELTY [51] | 8.00 | 6.50 | 0.25 | 1.25 |
| OBJCOVERAGE [47] | 18.50 | 10.75 | 0.75 | 0.50 |
| INTEXP | **27.25** | **27.00** | **3.75** | **19.00** |

**Figure 6: Success rates (%) and rewards on downstream tasks.** Knowing how to move (NOVELTY), or favoring objects (OBJCOVERAGE) is not sufficient to overcome sparse rewards in multi-step interaction tasks. Our INTEXP agent actively seeks out interactions to learn better policies in fewer training episodes.

## 4.2 Interaction exploration for downstream tasks

Next we fine-tune our interaction exploration agents for several downstream tasks. The tasks are (1) RETRIEVE: The agent must take any object out of a drawer/cabinet, and set it down in a visible location outside, (2) STORE: The agent must take any object from outside, put it away in a drawer/cabinet and close the door, (3) WASH: The agent must put any object inside the sink, and turn on the tap. (4) HEAT: The agent must put a pan/vessel on the stove-top, and turn on the burner.

These tasks have very sparse rewards, and require agents to successfully perform multiple interactions in sequence involving different objects. Similar tasks are studied in recent work [63, 55], which train imitation learning based agents on expert demonstrations, and report poor performance with pure RL based training [63]. Our idea is to leverage the agent's policy for intelligent exploration to jumpstart policy learning for the new task *without* human demonstrations.

We reward the agent (+10) for every subgoal it achieves towards the final task (e.g., for HEAT, these are "put object on stove", and "turn-on burner"). We fine-tune for 500k frames using PPO, and measure success rate over 400 randomized episodes from the same environments. The results in Fig. 6 (left) show the benefit of the proposed pretraining. Agents trained to be curious or cover more area (CURIOSITY and NOVELTY) are not equipped to seek out useful environment interactions, and suffer due to sparse rewards. OBJCOVERAGE benefits from being trained to visit objects, but falls short of our method, which strives for novel interactions. Our method outperforms others by large margins across all tasks, and it learns much faster than the best baseline (Fig. 6, right).

## 5 Conclusion

We proposed the task of "interaction exploration" and developed agents that can learn to efficiently act in new environments to prepare for downstream interaction tasks, while simultaneously building an internal model of object affordances. Future work could model more environment state in affordance prediction (e.g., what the agent is holding, or past interactions), and incorporate more complex policy architectures with spatial memory. This line of work is valuable for increasingly autonomous robots that can master new human-centric environments and provide assistance.

**Acknowledgments**: Thanks to Amy Bush, Kay Nettle, and the UT Systems Administration team for their help setting up experiments on the cluster. Thanks to Santhosh Ramakrishnan for helpful discussions. UT Austin is supported in part by ONR PECASE and DARPA L2M.

# 6 Broader Impact

Embodied agents that can explore environments in the absence of humans have broader applications in service robotics and assistive technology. Such robots could survey, and then give a quick rundown of a space for new users to, for example, alert them of appliances in a workspace, which of them are functional, and how these can be activated. It could also potentially warn users to *avoid* interaction with some objects if they are sharp, hot, or otherwise dangerous based on the robot's own interactions with them.

Deploying embodied agents in human spaces comes with challenges in safety — exploration agents than can "interact with everything" to discover functionality may inadvertently damage their environment or themselves, and privacy — navigating human-centric spaces requires agents to be sensitive of people and personal belongings. Careful consideration of these issues while designing embodied agent policies is essential for deploying these agents in the real world to collaborate with people.

## Footnotes

[1] We assume reliable odometry estimates. See Supp for experiments with noisy odometry.

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
