[Supplementary Material · supp.pdf]

## Supplementary Material

This section contains supplementary material to support the main paper text. The contents include:

- (§S1) A video comparing our INTEXP policy to other exploration methods on the interaction exploration task (Sec. 4.1) and supplementing qualitative policy results in Fig. 4.
- (§S2) Additional architecture and training details for our policy network and affordance models described in Sec. 3.3 in the main paper.
- (§S3) Implementation details for baseline exploration methods described in Sec. 4 (Baselines).
- (§S4) Environment-level breakdown of interaction exploration results presented in Fig. 3 in the main paper.
- (§S5) Additional ablation experiments for variants of our INTEXP reward formulation (Equation 1).
- (§S6) Interaction exploration experiments with noisy odometry.
- (§S7) Samples of affordance training data generated during exploration (Fig. 2, right panel) for marking schemes used by INTEXP(PT) and INTEXP(OBJ).
- (§S8) Additional affordance prediction results to supplement Fig. 5.

## S1  INTEXP policy demo video

The video can be found on the project page. In the video, we first illustrate the process of collecting affordance training data by placing *markers* during exploration (Sec. 3.2). We then show examples of our INTEXP agent performing interaction exploration on unseen test environments, along with comparable episodes by the other exploration agents presented in Sec. 4 of the main paper. In the video, the left panel shows the egocentric view of the agent at each time-step. The right panel shows the top-down view of the environment with the agent's trajectory and interaction attempts highlighted, similar to Fig. 4 in the main paper. We overlay predicted affordance maps on the egocentric view on the left panel for our method.

## S2  INTEXP architecture and training details

We provide additional architecture and training details to supplement information provided in Sec. 3.3 and Fig. 2 in the main paper.

**Policy Network**    For our policy network, we resize all inputs to $80 \times 80$. Each input is fed to a 3 layer CNN (conv-relu $\times$ 3) with channel dimensions $(N, 32, 64)$, kernel sizes $(8, 4, 3)$, and strides $(4, 2, 1)$ followed by a fully connected layer to reduce the dimension from $1152$ to $512$. $N$ is the number of input channels which is 3 for RGB frames, 1 for saliency maps and $|\mathcal{A}_I|$ for affordance maps.

Modalities are merged by concatenating and transforming using a fully connected layer back to 512-D, followed by an ELU non-linearity, following [12]. This representation is fed to a GRU with a single layer and hidden size 512, and then to two fully connected layers of the same dimension (512) to generate next action distribution and value. The network is trained using PPO, with learning rate 1e-4 for 1M frames, using rollouts of length 256 as mentioned in Sec. 3.3 in the main paper.

**Affordance network** $\mathcal{F}_A$    We use a publicly available implementation of the UNet architecture [49], built on a pretrained ResNet-18 backbone[1]. We make this network light-weight by using only 3 levels of encoder and decoder features, with decoder output dimensions $(128, 64, 32)$ respectively.

Inputs to this network are $80 \times 80$ RGB frames. As mentioned in Sec. 3.2, we train two segmentation heads, each implemented as a convolution layer on top of the UNet features. Both output $|\mathcal{A}_I| \times 80 \times 80$ affordance probability maps corresponding to $\hat{y}_A$ and $\hat{y}_I$ in Equation 3, respectively.

The network is trained with ADAM, for 20 epochs, with learning rate 1e-4. The learning rate is decayed to 1e-5 after 18 epochs. The cross entropy loss in Equation 3 is weighted by the inverse frequency of class labels present in the dataset to account for the label imbalance in the collected data.

**Training schedule details** As mentioned in Sec. 3.3, we train the policy network and affordance segmentation model iteratively. In practice, we train the policy network for 200k frames, then freeze the policy $\pi_E$ at this iteration to generate rollouts to collect our affordance dataset. Interleaving the affordance training more frequently results in better segmentation (+1.8% mAP) but no improvement in interaction exploration. Retraining models with reduced $M$=10k hurts performance (-2.8%), while larger $M$=500k results in similar coverage (+0.2%).

We collect a dataset of 20k frames as follows. For each (interaction, scene) pair, we maintain a heap of maximum size 200 which holds frames from a scene if they contain labeled pixels for that interaction, and we replace them if more densely labeled frames are encountered. This results in a more balanced dataset covering interactions and scenes, and translates to more robust affordance models.

## S3 Implementation details for baseline exploration policies

We present implementation details for each of the baseline exploration policies discussed in Sec. 4 (Baselines) in the main paper. All baselines use the same base architecture as our model described in Sec. S2, but vary in the reward they receive during training.

**CURIOSITY** Following [43], a forward dynamics model $F$ is trained to minimize the reconstruction error of future state predictions $\lambda_C||F(s_t, a_t) - s_{t+1}||_2^2$. The agent is rewarded a scaled version of this error in order to incentivize visiting states that it cannot predict well. We use $\lambda_C = 1e - 2$ and scale rewards by $0.5$. We explored ways to improve this baseline's performance, such as training it for $5\times$ longer and incorporating ImageNet pre-trained ResNet-18 features. However, while these improvements provided more stable training, they gave no better performance on the interaction exploration task.

**NOVELTY** Following [47], agents are rewarded according to $R(x, y) = \lambda_N / \sqrt{n(x, y)}$ where $n(.)$ counts how many times a location is visited. This incentivizes the agent to visit as many locations in the environment as possible. We use $\lambda_N = 0.1$.

**OBJCOVERAGE** Agents are rewarded for visiting new objects. "Visiting" an object involves being close to the object (less than $1.5m$ away), and having the object in the center of the image plane The latter criteria is implemented by creating a $60 \times 60$ box in the center of the agent's field of view, and checking if either 30% of the object's pixels are inside it, or 30% of the box pixels belong to that object. The reward function is the same form as Equation 1, but counting visits to objects as defined above, rather than successful interactions. This is similar to the criteria defined for the search task in [18] and the coverage reward in [47].

## S4 Environment-level breakdown of interaction exploration results

We present interaction exploration results for each unseen test environment in Fig. S1 to supplement the aggregate results presented in Fig. 3 in the main paper. On most environments, our INTEXP agents outperform all other exploration methods. Our agent performs best on FloorPlan1-2, which are the largest environments. FloorPlan4 is the smallest of the test environments (half the size of the largest) leading to all methods achieving higher coverage on average, though ours remains the best. OBJCOVERAGE eventually covers more interactions than our model on FloorPlan5, but is slower to accumulate rewards in earlier time-steps.

## S5 Additional ablation experiments for INTEXP variants

We present additional ablation experiments that investigate the form of the reward function in Equation 1. We compare the following forms:

- INTNOVELTY We reward the agent using a softer count-based reward $R = 1/\sqrt{n(a, o)}$, where $n(a, o)$ counts how many times an interaction is successfully executed. This reward incentivizes repeated interactions with the same objects, but offers more dense rewards.

**Figure S1: Environment level interaction exploration results.** Our agent outperforms other methods on large environments with multiple objects. It also discovers interactions faster on all environments.

**Figure S2: INTEXP reward ablations.** Explicitly training to maximize interactions using a hard count-based reward is more effective than softer reward variants that incentivize repeated interactions or ignore object instances.

- ACTNOVELTY We reward the agent for unique actions, regardless of object class $R = 1/\sqrt{n(a)}$. In this formulation, the agent is incentivized to broadly seek out more interaction actions, but potentially ignore object instances.
- INTEXP(RGB) The hard version of the reward presented in Equation 1 and Fig. 3 (right). This version only rewards the agent once per successfully discovered interaction.

All models are evaluated on coverage of interactions defined in Sec. 4.1 (Metrics). The results are presented in Fig. S2. The hard count-based reward proposed in Equation 1 is most effective in covering object interactions in new environments.

## S6 Interaction exploration experiments with noisy odometry

We inject truncated Gaussian noise into agent translation/rotation, which results in noisy marker positions, and thus noisy affordance training data. We select noise parameters similar to the LoCoBot platform (Murali et al.). Note, these errors compound over time. For translations we use ($\mu$=14mm, $\sigma$=5mm) and for rotations ($\mu$=1°, $\sigma$=0.5°), and truncated at 3 standard deviations. Our results in Fig. S4 show that our method outperforms other baselines that require odometry observations.

## S7 Affordance training data samples

As mentioned in Sec. 3.2, the agent collects partially labeled training samples to train our affordance model $\mathcal{F}_A$. We leverage two strategies to mark interaction targets: INTEXP(PT), which marks a single point at the center of the agents field of view, and labels pixels within a fixed distance around

**Figure S3: Affordance training data samples.** ( 🟩 success 🟥 failure ⬜ no-interaction ) Affordance labels generated during exploration using the two marking strategies presented in Sec. 3.2. **Top row:** markers are placed over entire objects. **Bottom row:** markers are placed at the exact target location, leading to fixed scale label regions. Note, in both cases, our models learn from incomplete (not all regions are interacted with) and noisy labels.

**Figure S4: INTEXP with noisy odometry.** Our method's advantages still stand against baselines that rely on noisy odometry observations.

this point, and INTEXP(OBJ) which uses true object boundaries from the simulator to label pixels of object instances at the target location. The latter corresponds to the agent understanding what visual boundaries are, without knowing object classes or affordances, which may be the output of a well trained foreground/background segmentation model. These strategies lead to different kinds of training data. See Fig. S3 for examples. Note that these affordance labels are noisy (many actions fail despite the action being possible under different circumstances) and sparse (all actions are not tried at every locations). Our model design in Sec. 3.2 accounts for this. See Sec. S1 for video demonstrations of this marking process.

## S8 Additional affordance prediction results

We show additional affordance prediction results in Fig. S5 to supplement the ones presented in Fig. 5 in the main paper. The last column shows failure cases that arise due to sparse and noisy labels. These include cases when objects are held in hand (which are rare in training data), and regions that are typically not interacted with (windows, curtains) which the model fails to ignore.

**Figure S5: Additional affordance prediction results.** ( ⬜ take ⬜ put ⬜ open ⬜ toggle ). See Fig. 5 in the main paper for more qualitative results, and Sec. 4.1 (Affordance predictions) for detailed discussion. Last column shows failure cases.

## Footnotes

[1] https://github.com/qubvel/segmentation_models.pytorch