[Reviews · NeurIPS 2020]

Review 1

Summary and Contributions: The paper proposes an approach where a robot explores an unknown environment in order to build a model of the affordances. The robot is performing a policy learned through reinforcement learning, where it is rewarded for maximizing successful novel interactions. At the same time the agent is learning an image segmentation model which is predicting the affordances of areas in the image. All the experimentation and validation was performed in the AI2-iTHOR environment.

Strengths: The authors correctly identify that learning the properties of an environment with exploration is an important feature of human learning, and it is one of the most promising approaches through which a robot can learn the affordances of the environment. As far as I can tell, learning a segmentation based model for affordances is novel.

Weaknesses: Equation 1: The authors do not discuss the fact that the way the reward function is written here, it is non-Markovian. It depends on the history of the state. It could be made Markovian if one folds the visitation frequency into the state, but then the equation (1) is not of the correct form. The approach essentially goes and tries every single object in the environment, and checks whether a certain action can be performed on it or not. It does not perform a fine grain differentiation between the actions - basically taking a knife or an apple is the same, and toggling the fireplace and the coffee makers are also the same action. Thus the number of affordances is very low. The paper does not really deal with the question of what is the impact on the environment if one tries every possible action on every possible object. Clearly, this looseness in the definition precludes any real world evaluation.

Correctness: As far as I can tell, the claims, method and evaluation approach are correct.

Clarity: Overall, the paper is well written.

Relation to Prior Work: I am not aware of a prior paper on the same subject.

Reproducibility: Yes

Additional Feedback: I read the authors feedback. I would like to point out that environment resetting techniques do not solve the problem of more informed exploration (especially when the exploration might involve doing things with a knife). There are certain things that simply cannot be learned by trying out actions. Overall, the feedback does not change my ranking.


Review 2

Summary and Contributions: The paper explores the important problem of learning affordances by interaction. Most previous works on learning affordances were based on manual annotations and passive approaches. In contrast, this paper explores an active approach in a dynamic environment to learn affordances. The paper proposes to learn an exploration policy and an affordance map jointly. This is a difficult search problem in the space of all objects, different types of affordances, agent locations, etc. The paper outperforms a number of baseline approaches and also provides ablation results. More interestingly, it shows the effect of pre-training using this method on a set of down-stream tasks.

Strengths: - The paper explores the interesting direction of learning affordances by interaction, which is a novel perspective compared to previous passive approaches. - The proposed approach has a been used as a pre-training step for a set of downstream tasks and shows improvement over alternative ways of pre-training. - The experiment section is comprehensive. It provides comparisons with a set of baseline approaches. It also provides a variety of ablation experiments. - The proposed approach outperforms the baselines in terms of precision and coverage metrics defined in the paper.

Weaknesses: - One of the main drawbacks of the paper is that it uses perfect odometry to compute the 3D world coordinates of the points. It would be much nicer if it used a noisy estimate of the odometry (using SLAM for example). It is interesting to see how the noise affects the results. - Some of the details are not clear: (a) In the IntExp(Obj) scenario, when the agent picks up the kettle, how does it know which pixels are pickupable? How does it know what the extent of the object is? (b) Lines 167-172 are not clear. It says "Classifier output y_A scores whether each interaction is successful at a location", while the condition for the indicator function is y=0 or y=1 (being either successful or unsuccessful). These are inconsistent. - It would be nice to provide the result of training with fully annotated images as an upper bound. I believe it is easy to obtain the annotations in THOR.

Correctness: Overall, the methodology seems correct.

Clarity: It is a well-written paper, but there are missing details (mentioned in the weaknesses section).

Relation to Prior Work: The paper does a good job of comparison with previous work.

Reproducibility: Yes

Additional Feedback: Comments after rebuttal: I read the other reviews and the rebuttal. The authors did a good job addressing the concerns. So I keep the initial 7 rating. I encourage the authors to include the new results in the revision.


Review 3

Summary and Contributions: Paper presents an exploration strategy for indoor embodied agents. Essentially it leverages an auxiliary 2D affordance map segmentation task on top of the main RL problem and feeds the predicted affordance map as an extra input of the policy network. Experiments on exploration in the AI-THOR simulator demonstrates its effectiveness over heuristic based counterparts on the proposed interaction coverage and precision metrics.

Strengths: + The paper is overall clearly written and easy to follow. + I can't find any technical issues within the main methodology. The proposed method is technically sound. + The baseline comparison are sufficient and covers a broad range of SOTA exploration methods, especially for embodied agents.

Weaknesses: - Some technical details deserves more elaborations, especially on the multi-task learning. The main idea of this paper is to train an RL agents simultaneously with an affordance map segmentation network, though these are essentially treated as two orthogonal objectives, the learning procedure in a whole is still unclear to me. The authors are suggested to provide more details on how the training is proceeded, such as whether these two tasks are trained concurrently or alternatively, and if their learning processes are not synchronous, how they choose the ratio of the learning iteration of each task, and how these extra parameters can affect the performances (in an extra ablation study). A comprehensive loss function and pseudo code will be preferred. - There is still some gap needed to be filled in the evaluation to further improve the sufficiency. To name a few: a) The selected metrics are only evaluated in a limited range. There are only curves over time(training steps) for the converge, while the success rates should also be evaluated in this way but not just the final quantities as it could be insightful to see the analysis on how the proposed method could improve the interaction skills. I think if it does perform as expected (also the counterparts), there should be low success rate at the beginning (as it tend to interact more with the object) but can improve faster than the other methods. b) The authors demonstrate how some downstream RL tasks could benefit from their proposed method and compare with the seemingly strongest baseline (obj coverage). Given the overall quality of their contribution, more evaluation efforts should be included here. I would like to see some endeavor on extending this part towards this direction: * To combine the proposed method with other exploration strategy. I do feel most of the considered baselines focus less on interaction but navigation, which seems sort of opposite to what this paper specifically works with, thus it may be more interesting to see some results on how the proposed method can really mitigate their drawbacks than simply contrasting on some interaction-oriented tasks. This can also further verify the main motivation of this paper---an efficient solution of exploration for interactions. Nevertheless, I do feel the selected downstream tasks could also be more challenging, say there is a significant need for both navigation and interaction.

Correctness: I can't find more than minor issues within the main methodology. The results can partly verify their claim on better interaction exploration, in the sense that some technical details on multi-task learning are missing, I cannot be fully confident on whether the comparisons are fair. The evaluation protocols are seemingly reasonable.

Clarity: The paper is clear and well written. I can't find any language issues.

Relation to Prior Work: I found the author did a good job on posing their method w.r.t. to prior work. Related papers are placed properly.

Reproducibility: No

Additional Feedback: I really find it confusing on why the point-based label can earn advantage over object-mask based label. As it shown in the left table of fig. 5, mask-based label delivers better affordance prediction but have a lower score on the proposed metrics. This can somehow contradict the main idea of this paper--the task of affordance map segmentation could improve the interaction exploration. The authors are expected to clarify on this. post-rebuttal === I've read the rebuttal and thanks the authors for the additional results and clarification.

[Author Response · NeurIPS 2020]


**Reviewer 1 [Score: 6]**

**Eqn 1: Reward depends on history of the state.** That's right. We can view the problem as a partially observable MDP (POMDP) where the state consists of the agent pose, interaction counts (visitation frequencies), object positions, etc. The recurrent policy network encodes the agent's observation history over time to arrive at a state-representation. Novelty rewards for visual exploration for mapping [57,51,7] are formulated similarly with RNNs.

**Approach tries every single object.** Actually, key to our approach is that our agents do *not* exhaustively try interactions — they learn to intelligently prioritize what to try (L43-57). The baselines that simply cycle through objects and actions yield very low precision (L264); our model discovers $2.5\times$ more interactions for the same time budget (L251).

**Taking a knife/apple is the same...# affordances is very low.** The reward in Eqn 1 is provided for every new *interaction* executed by the agent, where an interaction consists of an action (take, slice) coupled with an object instance (apple1, lettuce) — i.e. the reward *does* treat taking a knife vs. taking an apple differently. There are 103 total interactions (L203). This number is defined by the AI2-iTHOR environments and is in no way limited by our approach.

**Impact on environment if one tries every action/object.** As mentioned in Sec 6 (L309), safety is important to consider when developing interaction exploration policies in the real world. Methods that simultaneously learn to *reset* the environment (e.g., Eysenbach, ICLR 2018) are promising for enabling both safe and efficient RL in these scenarios.

**Reviewer 3 [Score: 7]**

**Noisy odometry.** If we add truncated Gaussian noise to odometry readings similar to the popular LoCoBot noise model (Murali et al.), we find that our method's advantages still stand against baselines that rely on odometry observations. See Fig. R1 (middle).

**How does INTEXP(OBJ) know object extents?** We use the true object boundaries from the simulator. This corresponds to the agent understanding what visual boundaries are, without knowing object classes or affordances (L176).

**Lines 167-172/Eqn 3 inconsistency:** $\mathcal{L}(\hat{y}_A, \hat{y}_I, y) = \mathcal{L}_{ce}(\hat{y}_A, y, \forall y_{ij} \neq -1) + \mathcal{L}_{ce}(\hat{y}_I, \mathbb{1}[y = -1], \forall y_{ij})$
Thanks for pointing this out. We have revised the equation above. Each loss term is evaluated over a subset of pixels (third argument) — e.g., for $y_A$, the classification is ($y = 1$ vs. $y = 0$), and is evaluated over pixels where ($y \neq -1$).

**Fully annotated images as an upper bound.** This upper bound is already reported in the paper — INTEXP(GT) in L226, yellow curve in Fig 3 (right). Fig. R1 (left) shows the equivalent upper bound for downstream tasks (Sec 4.2).

|                | Retr. | Store | Wash  | Heat  |
|----------------|-------|-------|-------|-------|
| INTEXP         | 27.25 | 27.00 | 3.75  | 19.00 |
| INTEXP(SAL)    | 25.00 | 13.25 | 3.50  | 16.50 |
| INTEXP+OBJCOV  | 23.00 | 11.75 | 4.50  | 20.25 |
| INTEXP(GT)     | 48.50 | 57.75 | 13.00 | 19.75 |

**Figure R1: Left:** Downstream success rates (%). **Mid:** Noisy odometry experiment. **Right:** Coverage vs. training epoch.

**Reviewer 4 [Score: 6]**

**Training schedule for policy/affordance learning.** We provide details in Sec S2 of the Supp. file. In short, we train the policy network for $M$=200k frames, then train the affordance model. Interleaving the affordance training more frequently results in better segmentation (+1.8% mAP) but no improvement in interaction exploration. Retraining models with reduced $M$=10k hurts performance (-2.8%), while larger $M$=500k results in similar coverage (+0.2%).

**Performance vs. training time, not just the final quantities.** Fig. R1 (right) shows coverage rate vs. training iters. The plot confirms R4's hypothesis. Note, Fig 6 (right) already showed a similar plot (reward vs. training epoch).

**How about extending...to combine another exploration strategy?** Fig. R1 (left) shows two such variants: (1) INTEXP (SAL)[1] does interaction exploration, but rather than affordance maps it uses saliency maps, which are also interaction/object-oriented; (2) INTEXP+OBJCOV combines our reward with object coverage rewards. It performs slightly worse on the interaction exploration task (-0.27%) but marginally better downstream on *wash* and *heat*.

**Why point-based $>$ obj-based?** It initially surprised us too. We found that the policy network tends to overfit more easily using the dense affordance predictions from INTEXP(OBJ) since the policy and affordance model use the same training environments. Its accuracy on training environments is indeed higher (25.16 vs. 23.61). The more conservative predictions from INTEXP(PT) generalize better to unseen test environments (Fig 3 right, L255-7).

## Footnotes

[1]We already showed INTEXP(SAL) in the submitted paper, Fig 3 (right); here we add it for the downstream tasks per R4's request.


[Meta-Review · NeurIPS 2020]

This paper has received 3 favorable reviews, indicating that the paper addresses an important problem (learning affordances from explorations and interactions) in a novel way through learning of segmentation model to the classical RL problem. The reviewers laud novelty, comprehensive evaluations, empirical results. The rebuttal could address several points raises by reviewers, e.g. noisy odometry. The strengths of the paper clearly outweigh the perceived weaknesses, and the AC concurs.